# Novel Tools to Measure Single Molecules Colocalization in Fluorescence Nanoscopy by Image Cross Correlation Spectroscopy

**DOI:** 10.3390/nano12040686

**Published:** 2022-02-18

**Authors:** Simone Pelicci, Laura Furia, Mirco Scanarini, Pier Giuseppe Pelicci, Luca Lanzanò, Mario Faretta

**Affiliations:** 1Department of Experimental Oncology, IEO, European Institute of Oncology IRCCS, 20139 Milan, Italy; simone.pelicci@ieo.it (S.P.); laura.furia@ieo.it (L.F.); mirco.scanarini@ieo.it (M.S.); piergiuseppe.pelicci@ieo.it (P.G.P.); 2Department of Oncology and Hemato-Oncology, University of Milan, 20122 Milan, Italy; 3Department of Physics and Astronomy “Ettore Majorana”, University of Catania, 95123 Catania, Italy; luca.lanzano@unict.it; 4Nanoscopy and NIC@IIT, CHT Erzelli, Istituto Italiano di Tecnologia, 16152 Genoa, Italy

**Keywords:** super resolution microscopy, colocalization analysis, Image Cross Correlation Spectroscopy, dSTORM, fluorescent nanodiamonds, correlative microscopy

## Abstract

Super Resolution Microscopy revolutionized the approach to the study of molecular interactions by providing new quantitative tools to describe the scale below 100 nanometers. Single Molecule Localization Microscopy (SMLM) reaches a spatial resolution less than 50 nm with a precision in calculating molecule coordinates between 10 and 20 nanometers. However new procedures are required to analyze data from the list of molecular coordinates created by SMLM. We propose new tools based on Image Cross Correlation Spectroscopy (ICCS) to quantify the colocalization of fluorescent signals at single molecule level. These analysis procedures have been inserted into an experimental pipeline to optimize the produced results. We show that Fluorescent NanoDiamonds targeted to an intracellular compartment can be employed (i) to correct spatial drift to maximize the localization precision and (ii) to register confocal and SMLM images in correlative multiresolution, multimodal imaging. We validated the ICCS based approach on defined biological control samples and showed its ability to quantitatively map area of interactions inside the cell. The produced results show that the ICCS analysis is an efficient tool to measure relative spatial distribution of different molecular species at the nanoscale.

## 1. Introduction

Physiological processes in biological systems are regulated by complex networks of interacting molecules with a precise spatial-temporal coordination. Optical fluorescence microscopy has proven to be a powerful tool for the study of these events, providing spatial maps of protein–protein interactions in the intracellular compartments. Colocalization is usually assessed by the detection of the overlapping fraction of two molecular species labelled by fluorescent antibodies [1,2]. However, the spatial resolution of widefield and confocal fluorescence microscopy is limited by diffraction to ∼200 nm, whereas molecular interactions occur at a much lower scale [3]. Several approaches have been implemented to bypass this resolution limit and achieve the nanoscale information. For instance, Forster Resonance Energy Transfer (FRET) [4,5,6] and in situ Proximity Ligation Assay (PLA) [7,8] are powerful methods to probe molecular distances in the nanometer range. Unfortunately, FRET is not sensitive when distances are larger than 10 nm [9] and quite hard to be efficiently implemented in immune-stained samples. PLA nanotechnology suffers of a low ligation and amplification efficiency, which may underestimate the number of interactions [10,11]. The development of Super-Resolution Microscopy (SRM) techniques (also called Nanoscopy), including Stimulated Emission Depletion Microscopy (STED) [12], Structured Illumination Microscopy (SIM) [13], and Single Molecule Localization Microscopy (SMLM), such as Photo-Activated Localization Microscopy (PALM) [14] and Stochastic Optical Reconstruction Microscopy (STORM) [15,16], broke the optical resolution barrier paving the way to the study of molecular interactions at the nanoscale level. Over the last decade, multicolor SMLM has revolutionized cell biology, enabling to investigate the relative organization and potential interactions between several molecules at a spatial resolution of about 20–50 nm or better inside the cell.

In SMLM fluorescent molecules switch between an active ‘ON’ or ‘bright’ state, where they emit fluorescent light when excited, and an inactive ‘OFF’ or ‘dark’ state in which they are switched off [17]. This phenomenon is observed in particular classes of synthetic fluorophores that reside for a prolonged time onto a metastable level (e.g., triplet state) before returning to the ground state. Particularly, in direct-STORM (dSTORM), switching probabilities can be modulated using fluorescent dyes that can photo-switch when immersed in selected buffers. Under laser irradiation, molecules are sparsely and stochastically switched-on allowing detection and fitting of Gaussian Point-Spread Functions (PSFs) for single molecule localization. Repeating this cycle over several thousand frames, all the localizations are accumulated to assemble a map of the molecular coordinates by using center-of-mass algorithms [18]. The resolution of the reconstructed single-molecule images is determined by the accuracy and precision of the individual molecules localizations, making dSTORM an effective approach for quantifying the distribution and colocalization of biomolecular species in cells.

Because SMLM images are obtained over thousands of time-lapse frames, a fundamental issue is represented by the correction of the sample drift introduced by the movements of the microscope stage [19]. Nanometer drifts can significantly compromise the precision of the calculated spatial positions, resulting in the blur of the reconstructed image or in the generation of artefactual features [20]. To correct for movements during image acquisition, stage drift is commonly estimated by tracking sequential localizations from fiducial markers, in combination with several drift-correction algorithms [21,22,23]. Generally, in optical imaging, different nanoprobes have been employed as fiducial markers, such as fluorescent and non-fluorescent beads, microfabricated patterns, gold nanoparticles, quantum dots, and DNA origami [24,25]. Multiple studies on the properties of nanomaterials compared the performances of these tools for drift correction. Considering parameters as brightness, photostability and phototoxicity, recent works demonstrated that fluorescent nanodiamonds (FNDs) are ideal fiducial markers [26,27]. FNDs contain nitrogen vacancy (NV) centers [28] and are biocompatible, are photostable under high-power laser excitation, have a high quantum efficiency, and have a wide excitation and emission bandwidth suitable for multicolor imaging. As shown by madSTORM [29], compared to fluorescent beads and nanogold particles, emission from FNDs was stable over time (4000 acquisitions at 100 ms/frame excited with a 647-nm laser (100 mW)), without emission intensity fluctuations, resulting in better localization precision and drift correction. As a result, FNDs are promising nanotools for single-molecule microscopy.

Even if SMLM approaches are now widely diffused for the description of intracellular structures at the nanoscale level, evaluating the correlation among spatial distribution of different molecular species is still under development and claims for new ad hoc analysis tools.

Methods of analysis of mutual colocalization in standard multicolor fluorescence microscopy can be traditionally divided into object-based methods and classical pixel-based methods [30]. SMLM instead produces map of the coordinates of the localized molecules. An object-based approach may consequently appear as the most natural one: molecules are immediately identified as objects by the list of XY-coordinates [31,32,33]. However, difficulties arise when considering that the localized fluorophores not necessarily reflect the position of different target biomolecules [29] introducing the need of a clusterization step to avoid redundant information. Moreover, the high amount of collected events can require extremely long times and high computational power.

On the other hand, classical pixel-based methods rely on the extraction of colocalization or correlation coefficients from each pixel intensity in the image (i.e., Pearson [34] and Manders [35] coefficients), making them versatile with every microscopy data set but not immediately transported to the localization maps produced by SMLM.

To quantify the colocalization of two labeled molecules, Image Cross-Correlation Spectroscopy (ICCS) has proven to be an attractive method [36]. ICCS corresponds to the spatial variant of fluorescence correlation spectroscopy (FCS) [37,38] and fluorescence cross-correlation spectroscopy (FCCS) [39], applying the same formalism to the analysis of the spatial intensity fluctuations between two signals. Notably, ICCS can be used as a pixel-based method, without pre-segmentation of the images. Oneto et al. [40] combined ICCS with dual-color STED nanoscopy extending the approach to the nanoscale. STED-ICCS provided a value for the colocalized fraction and the characteristic correlation distance associated to signal distributions. Moreover, the algorithm provides a local analysis that describes the spatial distribution of the measured correlation parameters at the intracellular level. STED-ICCS thus produced some of the most attractive features of an object-based analysis showing a spatial map of colocalization and the corresponding values of object-object distance. Recently, this method has been extended to Structured Illumination Microscopy (SIM) [41]. As demonstrated in STED-ICCS, the results showed that SIM-ICCS is a powerful method to measure nanoscale distances between two different particles.

Here, we extended the ICCS approach to SMLM by defining an experimental and computational pipeline for optimal data collection and analysis. We measured colocalization at single-molecule level and quantified the nanoscale distances between two molecular targets. Dual-color dSTORM acquisition was optimized to collect SMLM data for ICCS analysis first by correcting the spatial drift in all the observed channels. Thanks to their photophysical properties, FNDs have been employed as fiducial nanomarkers to calculate the stage drift during the single-molecule acquisition. Moreover, we employed ad-hoc strategies to target FNDs to different cell compartments according to the intracellular region of interest to grant an efficient correction of the drift on a wide range of experimental cases.

The produced results show that ICCS analysis is an efficient tool to measure the relative spatial distribution of different molecular species also at the nanoscale.

## 2. Materials and Methods

### 2.1. Cell Culture

MCF10A cells obtained from the American Tissue Culture Collection (ATCC) were grown in 50% DMEM High Glucose with stable L-glutamin (DMEM) (Euroclone) + 50% Ham’s F12 Medium (ThermoFisher Scientific, Waltham, MA, USA) containing 5% Horse serum, 50 ng/mL Penicillin/Streptomycin (both from Euroclone, Milan, Italy), Cholera Toxin (Merck Life Science, Milan, Italy.), 10 μg/mL Insulin (Merck Life Science, Milan, Italy.), 500 ng/mL Hydrocortisone (Merck Life Science, Milan, Italy.) and 20 ng/mL EGF (Pepro Tech, Cranbury, NJ, USA,) at 37 °C in 5% CO2. Cells were cultured on glass bottom dishes (MatTek, Ashland, MA, USA) coated with 0.5% (wt/vol) gelatin in PBS. Cells were grown for at least 48 h after seeding and fixed at 70–80% confluence to ensure a true exponential growth out of the initial lag phase.

### 2.2. Fluorescence Nanodiamonds (FNDs)

Nitrogen-vacancy-center fluorescent nanodiamonds of 40 nm size conjugated to streptavidin were acquired from Adamas Nanotechnologies (Adamas Nanotechnologies, Raleigh, NC, USA). The stock solution (in FBS with 0.1% BSA at 1 mg/mL (1% *w*/*v*)) was diluted 1:50 in DPBS 1X and added to the cells overnight at 4 °C.

### 2.3. Immunostaining

Fixed MCF10A cells were washed and permeabilized for 10 min in a permeabilization buffer containing 0.1% Triton X-100 (vol/vol) in PBS. Cells were then rinsed 3 times in PBS and incubated with primary antibodies in 5%BSA in PBS for 1 h at room temperature. The following primary antibodies were employed: Anti-Histone H3 3methylK9 (H3K9me3) (ab8898, Abcam, Cambridge, UK), Anti-BrdU (BD347580, BD Biosciences, Franklin Lakes, NJ, USA), Anti-Mre11 (ab12159, Abcam, Cambridge, UK), Anti-53BP1 (ab36823, Abcam, Cambridge, UK), anti phosphoH2A.X (ser39) (γH2A.X) (613402, Biolegend, San Diego, CA, USA), Anti-DYNLL1 (MAB2294, R&Dsystems, Minnneapolis, MN, USA). Cells were washed 3 times in PBS and incubated for 1 h at room temperature with the following secondary antibodies: Alexa Fluor^®^ 647 AffiniPure Goat Anti-Rabbit, (111-607-008, Jackson-immunoresearch, West Grove, PA, USA), Alexa Fluor^®^ 647 Goat Anti-Mouse IgG2b, (115-605-207, Jackson-immunoresearch, West Grove, PA, USA), Cy3 AffiniPure Goat Anti-Mouse IgG1 (115-165-205, Jackson-immunoresearch, West Grove, PA, USA) or CF™568 Goat Anti-Mouse IgG (H + L) (SAB4600312, Sigma-aldrich, St. Louis, MO, USA). For the positive control sample, MCF10A were first incubated with a DyLight 650 conjugated rabbit anti-53BP1_antibody (NB100-305C, Novus Biologicals, Centennial, CO, USA). We then incubated the same sample with an anti-Rabbit Cy3-conjugated secondary antibody. This way the distance between the two fluorophores is in the order of the size of the secondary-antibody molecule. For correlative analysis experiments, a biotin conjugated anti mouse antibody (68-36-011519, ThermoFisher Scientific, Waltham, MA, USA) was employed to mark γH2A.X antibody, followed by Streptavidin Atto-425 (800-656-7625, Rockland Immunochemicals, Limerick, PA, USA). Cells were again rinsed 3 times in PBS and briefly re-fixed in 4% paraformaldehyde (wt/vol) for 5 min. After the last wash in PBS, dSTORM imaging buffer was added to the samples.

### 2.4. dStorm Imaging Buffer

Single-molecule dSTORM imaging was performed in an imaging buffer that included Buffer A (10 mM Tris (pH 8.0) + 50 mM NaCl), an oxygen-scavenging system “GLOX” (56 mg/mL Glucose Oxidase (Sigma-aldrich, St. Louis, MO, USA), 3.4 mg/mL Catalase (Sigma-aldrich, St. Louis, MO, USA) Stock (17 mg/mL Catalase in dH20)), and MEA (1 M Mercaptoethylamine (Sigma-aldrich, St. Louis, MO, USA)) Stock (77 mg MEA + 1.0 mL 0.25 N HCl). MEA solution was kept at −20 °C and used within 2–3 weeks of preparation.

### 2.5. Confocal Imaging

Correlative data were acquired with a commercial inverted two-layer Nikon Eclipse Ti2 microscope, equipped with an A1R confocal scanhead and N-STORM imaging module (Nikon instruments, Tokyo, Japan) and controlled by the NIS Elements software (version 5.30.02). A Nikon CFI SR Apochromat TIRF 100× oil, 1.49 NA objective was used for all the measurements. Confocal images were scanned in galvanometric mode and the ROI size was set to 1024 × 1024 pixels, with a pixel size of ∼0.07 μm. The scanning pixel dwell-time and laser power were set to avoid the photobleaching effect. Confocal and widefield (dSTORM) images were aligned targeting the position of FNDs using the alignment routine of NIS Elements software (version 5.30.03, Nikon instruments, Tokyo, Japan) (Pelicci et al., in preparation).

### 2.6. dSTORM Imaging

Single-molecule imaging was performed with a super-resolution Nikon N-STORM microscope configured for oblique incidence excitation (N-STORM module 2, Nikon instruments, Tokyo, Japan). dSTORM acquisitions were performed in STORM imaging buffer, described above. Alexa Fluor 647 and DyLight650 dyes were excited by using 647 nm laser (120 mWatts nominal power), while Cy3 and CF568 were excited with a 561 nm laser (70 mWatts) (LU-NV laser unit, Nikon instruments, Tokyo, Japan). A 405 nm laser (20 mWatts) was used for continuous activation of dyes (both the activator and imaging lasers are continuously on). A multi-band dichroic mirror (C-NSTORM QUAD 405/488/561/647 FILTER SET; Chroma), combined with 561 nm- and 64 7 nm- filter cubes (IDEX Health & Science, Semrock Brightline^®^, West Henrietta, NY, USA), was used to filter the fluorescence excitation and emission. This filter-set avoided the crosstalk between channels, also blocking fluorescence due to the 405 nm-activator laser. During the 2D dual-color imaging, fluorescent nanodiamonds (FNDs) of 40 nm size conjugated to Streptavidin were imaged in both acquisition channels (647 nm and 561 nm). The fluorescence emission of all channels was collected through a Nikon CFI SR Apochromat TIRF 100× oil objective (1.49 NA) and finally detected by an ORCA Flash 4.0 Digital CMOS camera C13440 (Hamamatsu, Bridgewater, NJ, USA). The number of frames and the exposure time per channel depend on the density pattern of the immunostaining and on the dye blinking-state. In each acquisition, we recorded 15.000 frames at 20 ms/frame of exposure time per channel. The selected z-plane position was maintained by monitoring the reflection of a near infrared light from the coverslip inner surface (Nikon Perfect Focus System (PFS)). The employed fraction of the full power of both activation and excitation lasers is dependent on the blinking efficiency of the single fluorophores. FNDs were detected every 1000 frames by selective excitation at 488 nm. Single Molecule Localization fitting was performed with Offline N-STORM Analysis module (NIS Elements software version 5.30.03, Nikon instruments, Tokyo, Japan). Drift was corrected calculating the shift of the FNDs positions between frames in every channel. FNDs were characterized by complete absence of blinking: their persistence in the image and localizations, as evidenced by 488 nm selective excitation, allowed to clearly identify them. Events from the NDs containing regions were then excluded from the analysis (see Appendix A). After the localization analysis, the reconstructed STORM images are generated. In the super-resolution image reconstruction, each molecule is represented by a Gaussian spot localized by the centroid position, with the localization precision obtained from the single-molecule fit and by an amplitude value related to the number of emitted photons.

### 2.7. Colocalization Analysis

Image cross-correlation spectroscopy (ICCS), for nanoscale colocalization analysis, was performed with the open-source code written in Matlab (The Mathworks, Natick, MA, USA) and available at https://github.com/llanzano/ICCS (accessed on 1 September 2020). For details on the software refer to Oneto et al. [40]. Briefly, images cross correlation and auto correlation functions were obtained by averaging over the pixels contained in a selected region of interest (e.g., cell nucleus). The amplitude parameters from the resulting curves were employed to calculate two coefficients of localizations whose arithmetic mean provides the colocalizing fraction *f_ICCS_*. The measured widths are instead related to the broadening of the cross-correlation function that is a parameter sensitive only to the distance *d* between correlated particles. Details on the analysis can be found in the Appendix A.

As reported in Oneto et al. [5], local cross correlation is iteratively calculated on small square 69 × 69 pixels wide subregions of the full-size image.

## 3. Results

### 3.1. Streptavidin Conjugated Nano-Diamonds as Intracellular Reference Markers

SMLM, such as dSTORM, is based on detection and localization of single molecular fluorescent-blinking events. Images are obtained over thousands of time-lapse frames (1000–100,000 frames at a speed of 10–1000 frames/s, taking several minutes or longer), achieving less than 50 nm of lateral resolution [42,43]. Owing to the long acquisition time, nanoscale movements of the microscope stage can significantly compromise the precision of the final super-resolved image, reducing the resolution and generating artifacts in the reconstructed images [20]. In a previous work [29] FNDs have been employed as reference to correct spatial drift during long STORM acquisitions. They were deposited on a glass coverslip and observed by the evanescent field created under total internal reflection. However, such a solution makes their use poorly compatible with an analysis that targets different intracellular compartments, e.g., the cell nucleus. We thus verified the photophysical properties of FNDs conjugated to streptavidin. After overnight incubation, a streptavidin-ND complex can be targeted to the mitochondria where high amounts of endogenous biotin are physiologically accumulated (Figure 1 panel a, b and inset c). Due to their wide excitation and emission spectra and to their high brightness, intracellular FNDs can be excited with blue laser light and observed both in the yellow-orange (Figure 1 panels a and b, STORM channel 2) and in the far red region (Figure 1 panels a and b, STORM channel 1) of the visible spectrum and isolated from the surrounding fluorophores. This way, the position of their center of mass allows calculating and efficiently correcting the spatial drift of the localized single molecules (Figure 1 panel c).

Fluorescence emission from streptavidin conjugated FNDs remains stable for the entire acquisition period confirming the complete absence of photobleaching (Figure 1 panel d,f) even in presence of the buffer employed to induce the blinking effect typical of dSTORM.

Finally, we verified that the photostability of FNDs was maintained even under localized high-power laser fluxes as the ones employed in confocal imaging. The possibility to observe FNDs in confocal imaging makes them a good reference marker for registration in correlative confocal-STORM microscopy as detailed below (see Section 3.3).

### 3.2. Image Cross-Correlation Spectroscopy to Evaluate Colocalization of Molecular Species in SMLM

SMLM calls for specific procedures to analyze the relative distribution of spatial coordinates calculated for two different biomolecular species. ICCS has been successfully employed to evaluate colocalization at the nanoscale in super-resolved images (Oneto et al. [40]).

ICCS is based on Fourier analysis of conventionally formed images. SMLM provides instead a list of coordinates of molecules. To reconcile the two approaches, we choose to re-image the calculated molecular coordinates as a superposition of Gaussian Point Spread Functions (PSFs) of variable width.

Figure 2 reported examples of reconstructed images with a Gaussian width of 10 and 50 nm. We opt for the higher value considering that single events collected during STORM acquisitions represent the position of the reporter fluorophores conjugated to a secondary antibody and that the same emitter can be switched-on more than once during the acquisition. Single localized events do not necessarily correspond to different protein molecules, but they just report the position of the secondary antibody creating a problem of degeneration (multiple localized events for the same target molecule). A 50 nm-width PSF can efficiently represent the molecular tree formed by the primary and secondary antibody complex on the target protein avoiding the degeneration introduced by multiple emissions. The chosen value is also close to the typical resolution calculated for SMLM super resolved images.

In ICCS, the shape of the cross-correlation function (amplitude and width) depends on the spatial resolution of super-resolved images and the distance *d* between the two molecules of interest in the two channels. The fraction of correlated molecules *f_ICCS_*, corresponding to the amplitude of cross-correlation curve, is extracted by the corresponding parameters of single 2D auto-correlation functions of the two channels, also providing information about the average distance between correlated molecules.

To validate the ICCS as a tool to evaluate colocalization, we initially analyzed a positive and a negative control sample. Representative dual-color dSTORM images of diploid mammary epithelial MCF10A cells are reported in Figure 2. Each image shows (i) the image auto- and cross-correlation curves, (ii) the mean value of colocalized fraction *f_ICCS_*, (iii) the mean distance value *d* extracted by ICCS, and (iv) the map of the colocalized fraction obtained by local ICCS.

As positive control, we marked the same protein with different fluorophores to produce two highly correlated molecular distributions (Figure 2 panel a). The high value calculated for the colocalized fraction (*f*_ICCS_ = 0.52 ± 0.06, mean ± SD, n = 11 cells) is compatible with the adopted immunostaining approach. The mean distance *d*_53BP1-53BP1_ = 32 ± 6 nm (mean ± SD) is close to the chosen spatial width of single molecules while the correlation coefficients across the nucleus are homogenously distributed with local peaks corresponding to regions of protein accumulation. To evaluate the strength of the obtained numbers, it is necessary to remember that the efficiency of localizations for each molecular species strongly depends on many factors, e.g., the blinking efficiency of the reporter fluorophore. Even if without a precise calibration to measure the number of fluorophores per antibody molecule and a titration of the number of antibody molecules per protein target, the estimated value represents a good reference point for qualitative colocalization analysis.

The colocalized fraction (*f**_ICCS_* = 0.07 ± 0.02, mean ± SD, *n* = 5 cells) and the distance value (*d*_H3K9me3-BrU_ = 105 ± 16 nm, mean ± SD) calculated on the cell nuclei of the negative control sample are in complete agreement with the chosen biological model: tri-methylated lysine 9 of histone H3 (H3K9me3) is a marker of heterochromatin where the level of transcription (marked by incorporated nucleoside Bromo-Uridine) are close to zero (Figure 2 panel b).

### 3.3. Correlative Microscopy and Image Cross-Correlation Spectroscopy as a Tool to Measure Compartmentalization of Biomolecular Interactions

Super resolution microscopy, and in particular SMLM, is often limited by the relatively low number of simultaneously collectable channels. dSTORM employs selected fluorophores with homogenous photophysical properties and a proper set of high-power lasers to excite them, with the consequent request for a dedicated and expensive setup. A modulation of the required spatial resolution according to the observed fluorescent molecules allows to bypass this limitation coupling diffraction-limited and super-resolved imaging modalities. According to this concept, we performed a correlative analysis of confocal images and SMLM localization data employing FNDs as markers for spatial registration.

Figure 3 shows an example of such an approach. 53BP1 and MRE11 are members of the DNA Damage Response (DDR) molecular machinery. ICCS analysis performed on the entire cell nucleus revealed a colocalized fraction of *f**_ICCS_* = 0.19 ± 0.03 (mean ± SD, *n* = 6 cells) and a value of the mean distance d *d*_53BP1-MRE11_ = 61 ± 16 nm (mean ± SD).

Both the proteins are functionally related to the recognition of a breakage in the genome and to the initiation of the cascade of events leading to the accumulation of the phosphorylated form of the H2A.X histone variant (γH2A.X) and recruitment of the DDR machinery.

We thus decided to evaluate the colocalization degree of 53BP1 and MRE11 proteins in proximity of DNA double strand breaks, as an applicative example of the correlative analysis between SMLM and confocal microscopy. γH2A.X foci were thus detected by diffraction limited confocal microscopy to localize the damaged DNA inside the nucleus. Dual color dSTORM series of images were then collected to evaluate distribution of the 53BP1-MRE11 proteins of interest and the resulting data registered with the acquired confocal images. ICCS analysis was finally applied in a region created by segmenting γH2AX foci and compared to the results previously calculated over the entire nucleus. Data show increased colocalization parameters with a colocalized fraction of *f**_ICCS_* = 0.35± 0.11 (mean ± SD, *n* = 6 cells) and a value of the mean distance *d*_53BP1-MRE11_ = 71 ± 33 nm (mean ± SD) overlapping the one measured on the entire cell nucleus. Correlative 3 color confocal-STORM analysis thus allowed unmasking the expected simultaneous recruitment to DNA double strand breaks of 53BP1 and MRE11 in response to genomic damage, showing at the same time that their interaction is not spatially confined to this site in the nucleus.

## 4. Conclusions

Overcoming the diffraction barrier made optical fluorescence microscopy an efficient tool to investigate the scale below a hundred nanometers. New analysis procedures are consequently required to provide a better description of the world of biomolecular complexes. At the same time, advanced microscopy can benefit from a growing number of solutions created by nanotechnologies.

In this work, we presented an experimental pipeline (summarized in the Appendix A in the Appendix A) that optimizes the acquisition and analysis of multicolor SMLM data for biomedical research to evaluate interactions between biomolecules.

SMLM data collection can be degraded by potential artifacts due to the extremely high number of acquired images, and consequently to the long acquisition time. Optimizing the correction of spatial drift is instrumental to achieve high localization precision. We showed that targeting fluorescent nanodiamonds (FNDs) to an intracellular compartment (e.g., mitochondria), by conjugating them to streptavidin, provided an efficient solution to eliminate spatial drift in dual color STORM acquisitions. Differently from beads or FNDs deposited on glass coverslip or cells, the Streptavidin-FND complexes can be easily retrieved in all cells and at every z plane. The choice of the target cell can be entirely based on the phenotype of interest without any dependence on the presence of the reference tools required for drift compensation. The same approach allows targeting the FNDs to other cell compartments (Pelicci et al., in preparation). This way FNDs can be localized to a biotinylated cell membrane leaving the cytoplasm free for measurement of the molecules of interest.

To evaluate the relative spatial distribution of the molecular species under investigation, we extended Image Cross-Correlation Spectroscopy (ICCS) analysis to SMLM to measure the colocalization degree in dual-color dSTORM images. In SMLM (PALM/STORM), object-based methods are considered the most suitable tools to measure molecules colocalization. Clustering algorithms, such as distance to nearest-neighbor [44], DBSCAN [45], and Ripley’s K function [46], are defined as the reference methods, where the molecules (objects) are first segmented and then represented as points through coordinates of their mass center in the field of view. However, most of the clustering algorithms depends on the assignment of parameters and, due to the high number of detected objects, require high computational power and/or long processing times. Finally, the performances of object-based approaches generally decrease in very crowded environments [40] with highly dense signals as in the case of chromatin and transcription factors in the cell nucleus. More than this, SMLM, by definition, is able to locate single events inside areas of dense signal. However, it suffers limitations related to the indirect detection of the targeted molecule via primary and secondary antibodies. Degeneration or redundancy of events is an intrinsic problem of localization microscopy: separated events do not necessarily represent different target proteins or even different antibody molecules since the same fluorophore can be detected more than once during the acquisition period or can be bound to the same antibody.

We showed that ICCS analysis adapted to SMLM also provides an efficient tool to measure the colocalization level of multicolor STORM data thanks to the advantage of a pixel-based approach. The passage from a particle-coordinates list to a pixel view is obtained by the superposition of Gaussian PSFs centered on each localized event, thus bypassing the problem of intrinsic redundancy. In addition, ICCS is parameter free.

A first important technical aspect to consider is that the computational speed of ICCS depends on the number of pixels but not on the number of objects in the image (as in object-based methods). Nonetheless, we noted that ICCS algorithm works in a reasonable time (<1 min/cell) also with large size images (format 2000 × 2000 pixels, pixel size 10 nm/pixel) with standard RAM computing, while managing hundreds of thousands of objects would require video cards programming to process data in a reasonable time. An apparent limitation of ICCS is that it provides only an average description of the properties of the sample in the analyzed region. However, the presented method can be be spatially scaled down to provide a local quantification of the colocalizing molecules inside the cell. By testing selected reference biological samples with different degree of correlation, we confirmed dSTORM-ICCS is able to efficiently detect the colocalized fraction and the relative correlation distances associated to the targeted particles.

However, the functional description of specific hotspots in the colocalization maps can be limited by the size of the targeted area, to obtain an adequate statistical sampling of pixels, and by the reduced number of channels that can be simultaneously acquired in STORM microscopy. We therefore introduced a correlative dSTORM-confocal microscopy analysis able to collect three or more channels at variable resolution taking advantage of the photostability of FNDs. FNDs do not suffer of any photobleaching effect even under the high-density photon fluxes employed in confocal imaging, thus providing a reliable registration marker. Intracellular compartments can be consequently localized and imaged at a diffraction limited scale to drive a single-molecule analysis of the colocalization confined to the region of interest. We demonstrated the potential of such an approach by monitoring the interaction between two components of the DDR machinery in proximity of DNA Double Strand Breaks revealing the differences in the amount of colocalizing 53BP1-MRE11 molecules inside and outside γH2A.X foci. The increase in the calculated colocalization parameters reflects the interaction of the two proteins during activation of the DDR machinery in proximity of DNA double strand breaks.

In summary, we extended the panel of available tools to collect and analyze data for advanced microscopy applications in biomedical basic research. A pixel-based imaging approach was adapted to analyze maps of molecular coordinates enlarging the range of ICCS applications in super resolution microscopy from STED and SIM to multicolor SMLM and providing a validated, easy to use and versatile approach to the quantitative description of molecular interactions. ICCS analysis does not intend to replace other well performing object-based algorithms, but it can be considered as a complementary tool to help in localizing molecular interactions. A first rapid ICCS-based evaluation of the correlation among signals can be employed to identify potentially highly correlated regions. Then, a more detailed clustering-driven object-based analysis can be applied to targeted areas calculating the molecular coordinates on the ten of nanometers scale, thus optimizing computation times and resources.

## Figures and Tables

**Figure 1 nanomaterials-12-00686-f001:**
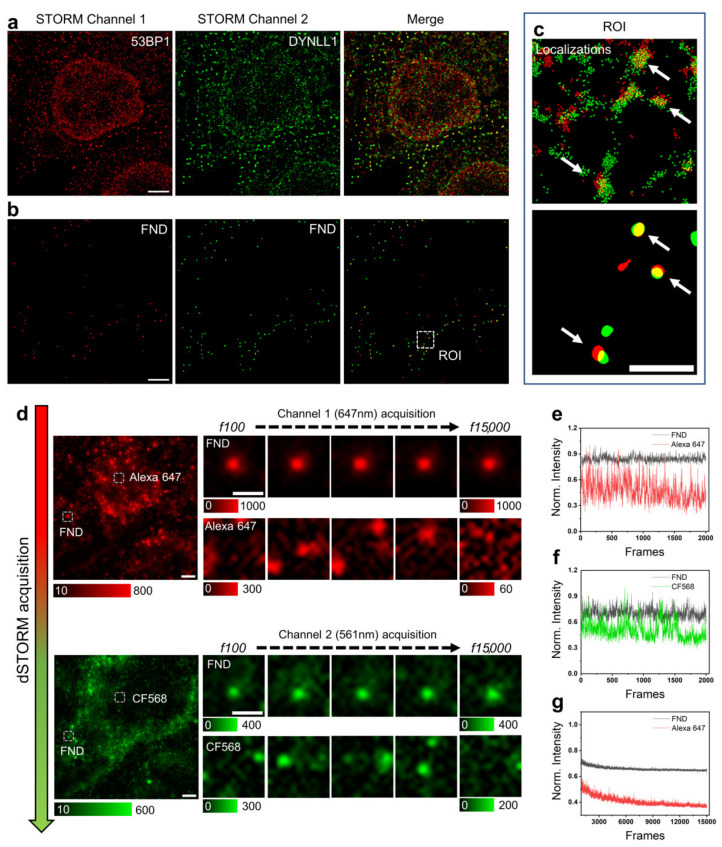
Dual-color dSTORM acquisition workflow with fluorescence nanodiamonds (FNDs). (**a**) Representative dSTORM images of MCF10A cells upon targeting of 53BP1 (red) and DYNLL1 (green), stained with AlexaFluor647 and CF568, respectively. Shown are (from left to right) the single-color 53BP1 and DYNLL1 images and the dual-color dSTORM image. (**b**) FNDs spots positions obtained from red and green STORM channels. (**c**) Zoomed region of the highlighted area (white dashed box) in (**b**). The upper panel shows the localization events (each colored pixel represents one localized single-molecule event) recorded during the acquisition. Representative FNDs are indicated by the arrows. In the lower panel, the recorded events were transformed into Gaussian PSFs of fixed width and segmented (see Materials and Methods and Appendix A for details) to highlight the drift correction (Scale bar ROI: 1 μm) (**d**) Representative images of dual-color dSTORM acquisition. The fluorophore blinking in timelapse imaging (15,000 frames) for each channel (53BP1-AlexaFluor647 and DYNLL1-CF568) is shown. The bright spots in the image correspond to the fluorescence emission of individual fluorophores. Most of the fluorescent labels are switched off such that the fluorescent molecules are well separated. FNDs show no blinking properties, indicating high photostability in both channels. (**e**,**f**) Comparison of the photon emission from fluorescent nanodiamonds (FND, 40 nm; black), Alexa Fluor 647 dye (red) and CF568 dye (green) over 2000 acquired frames using 561 nm and 647 nm laser excitations, measured in the highlighted regions (white dashed box). (**g**) Fluorescence decrease of the 40 nm FNDs (black) and the AlexaFluor647 (red), over 15,000 frames of image. Scale bar: 3 μm.

**Figure 2 nanomaterials-12-00686-f002:**
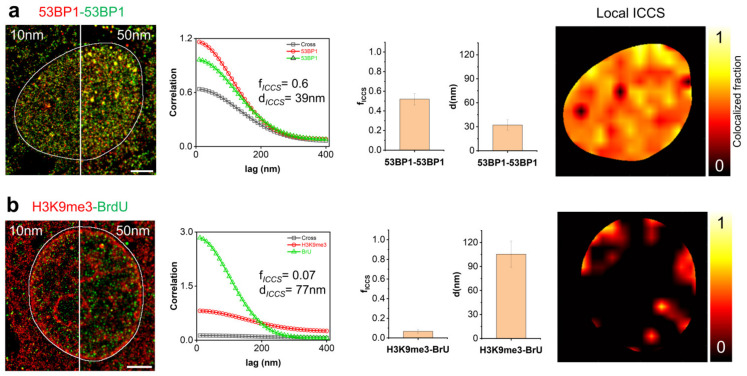
Analysis of single molecule colocalization by ICCS. Analysis of representative dSTORM images of MCF10A nuclei acquired upon labeling of (**a**) 53BP1 (green) and 53BP1 (red), (**b**) BrU (green) and H3K9me3 (red). Shown are (from left to right) the dual-color STORM image at different spatial resolution (10 nm and 50 nm), the spatial correlation functions recovered by ICCS, the colocalized fraction (*f_ICCS_*) and the values of distances (*d*) extracted from ICCS analysis (data are mean ± s.d. of the mean values of *f_ICCS_* and d calculated on each nucleus (delimited by the white line) over different cells (see text)), and the map of the colocalized fraction recovered by local ICCS. The ICCS plot shows the cross-correlation function (black squares) and the red (red circles) and green (green triangles) channel autocorrelation functions along with the corresponding fits (solid lines). Scale bar: 3 μm.

**Figure 3 nanomaterials-12-00686-f003:**
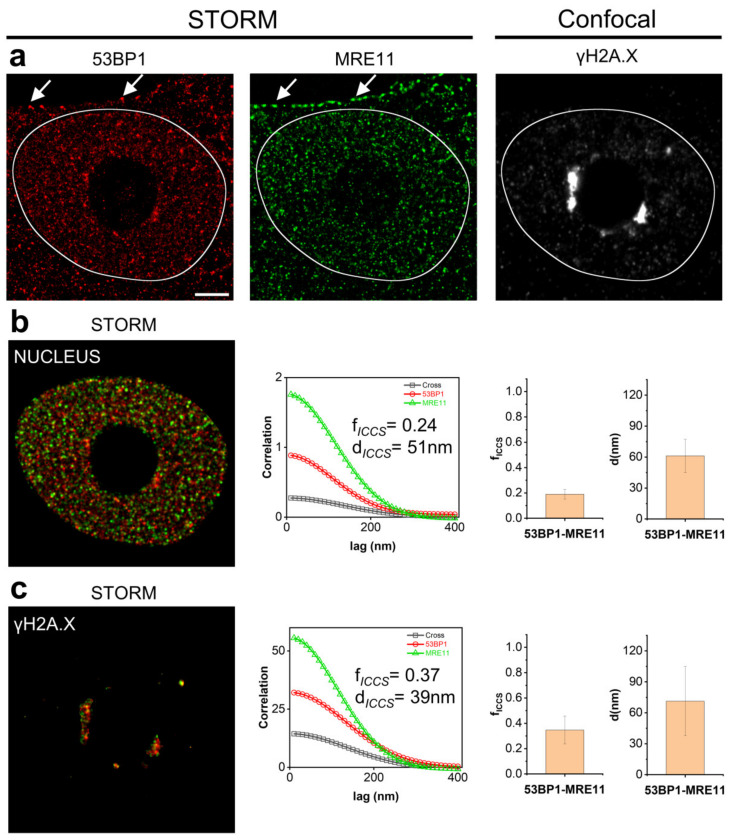
Comparison of 53BP1 and MRE11 colocalization in different nuclear areas by correlative microscopy driven ICCS analysis. (**a**) Correlative dSTORM-Confocal imaging: dSTORM images at single molecule resolution of 53BP1 (red), MRE11 (green), stained with DyLight 650 and Cy3, respectively, and diffraction-limited confocal image of γH2A.X labeled with Atto425. White arrows show FNDs spots in both the STORM channels localized on the cell plasma membrane. (**b**) Mask of nuclear area of MCF10A cell. Shown are (from left to right) spatial correlation functions recovered by ICCS and colocalized fraction (*f_ICCS_*) and the values of distances (*d*) extracted from ICCS analysis restricted to the masked area (data are mean ± s.d. of the mean values of *f_ICCS_* and d calculated on each cell). (**c**) Mask generated by γH2A.X intensity signal to discriminate DNA damage accumulation regions. Shown are (from left to right) spatial correlation functions recovered by ICCS, and colocalized fraction (*f_ICCS_*) and the values of distances (d) extracted from ICCS analysis restricted to the masked area (data are mean ± s.d. of the mean values of *f_ICCS_* and *d* calculated on each cell). Scale bar: 3 μm.

## Data Availability

Raw Data are available upon request.

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
