# Peer review of "Novel Tools to Measure Single Molecules Colocalization in Fluorescence Nanoscopy by Image Cross Correlation Spectroscopy"

_nanomaterials, 2022, doi:10.3390/nano12040686_

Round 1

Reviewer 1 Report

see letter attached

Reviewer 2 Report

This manuscript describes the application of ICS and ICCS for the quantitative analysis of colocalization in images acquired via STORM, similarly to what has previously done for STED. 

The work is interesting and well written but there are some details that should be improved. Many explanations regarding the details of the experimental pipeline are missing.

Major points:

It should be better explained how ND are identified and distinguished from other signal. The differences between the two types of signal are clearly described but how were they used in the experimental pipeline exactly? How is panel b in Fig. 1 obtained concretely? Thresholding?

Not clear if alternated excitation/detection scheme is used (as hinted by the arrow in fig.1 d). In that case, how can one identify ND in images and distinguish them from 100% colocalized proteins?

I assume ND signal has to be excluded before proceeding with ICCS analysis. But very few details are given in general regarding the specifics of the analysis. 

Line 282; It is not clear how the amplitude is chosen. 

Line 306; not clear how f_ICCS was calculated. Is it normalized to the green or red AC? And why is it compatible with the adopted approach? Assuming perfect and completely random labelling (RG, RR, GG, GR), the 2 AC should be similar and the CC should be ca. 1/3 of the AC. I hope I am not wrong with my calculations (I considered that the non correlating particles have a double brightness in each channel).

Line 310; is actually the colocalized fraction actually changing with higher C? (not an issue if the shown fraction is indeed normalized with the AC)

How are ICCS maps obtained? I assume with image-subsets? How many pixels? What shape? 

How are AC and CC curves obtained from the corresponding 2-d surfaces? By averaging radially?

The width of both AC curves should be 50 nm. Is this actually the case? And how does this intrinsic width factor in the calculation of the d_ICCS?

Line 322; what does the d of 105 nm tell us? Shouldn´t it depend on the actual protein density and therefore changing strongly from cell to cell?

Connected to the previous point, line 407, a test to verify the meaning of the relative distance is not presented. A control in this sense would be helpful.

Minor points:

Line145; not clear how cells were fixed "to guarantee exponential growth"

Laser power densities should be specified.

Line 235; not clear how ND adjacent to an observed cells is poorly compatible with intracellular imaging.

Line 289: one additional line to explain "the problem of degeneration" would be helpful.

Round 2

Reviewer 1 Report

The authors have considerably improved the quality of the manuscript, especially by adding the supplement more details are available for better  understanding. The authors addressed all points of concerns with clear explanations and improved article text passages.

The present form of the manuscript can by published by Nanomaterials.

Author Response

We thank the reviewer for the comment of our work

Reviewer 2 Report

The authors have addressed all the issues that were raised in the previous review round. Just a small note on my previous point n.5 regarding lines 330-1 (revised version). This was misunderstood and I apologize for that. The revised version presents a more detailed explanation of the labelling procedure of the positive control sample. The secondary green antibody should mostly bind to the red primary antibody, so I would expect few red-only dots, even fewer green-only dots and many green-red dots. In line 331 the authors mention that the f_ICCS of 0.52 is compatible with the labelling procedure, thus suggesting that this value can be indeed be used for a quantitative analysis. Maybe the authors can specify why a value of 0.52 is expected. According to my calculations (considering AC_R=1.2, AC_G=1 and f_ICCS=0.5), the number suggests ca. 24% red-only dots, 38% green-only dots and 38% green-red dots. 
Maybe one more sentence explaning how/if the f_ICCS can be used for a quantitative estimation of complexes (even maybe referring the older paper) would be helpful for the readers. 

Author Response

Just a small note on my previous point n.5 regarding lines 330-1 (revised version). This was misunderstood and I apologize for that. The revised version presents a more detailed explanation of the labelling procedure of the positive control sample. The secondary green antibody should mostly bind to the red primary antibody, so I would expect few red-only dots, even fewer green-only dots and many green-red dots. In line 331 the authors mention that the f_ICCS of 0.52 is compatible with the labelling procedure, thus suggesting that this value can be indeed be used for a quantitative analysis. Maybe the authors can specify why a value of 0.52 is expected. According to my calculations (considering AC_R=1.2, AC_G=1 and f_ICCS=0.5), the number suggests ca. 24% red-only dots, 38% green-only dots and 38% green-red dots. 
Maybe one more sentence explaning how/if the f_ICCS can be used for a quantitative estimation of complexes (even maybe referring the older paper) would be helpful for the readers. 

We thank the reviewer for the comment. Due to the high number of experimental variables inside a dSTORM acquisition, a quantitative analysis should require additional complex procedures to measure the number of fluorochromes per antibody molecule and a titration of the number of antibody molecules per target (i.e. primary antibody versus protein target; secondary antibody versus primary antibody). The different blinking efficiency can consequently be a reason for the discrepancies in the red-green ratio noticed by the reviewer. A647 is one of the best fluorophores available for dSTORM analysis and in our set up the 647 nm laser is the most powerful one. Detection of molecules in the red channel  is consequently easier and probably most efficient even if associated with a directly conjugated antibody. Moreover the use of a secondary antibody (with more than one bound antibody per target molecule) can be paradoxically less efficient due to the degeneration problem, leading to less events associated to a single point. In the same time, isolated events can be recorded due to the increased biological background typical of the secondary antibody based amplification of the signal. Based on these considerations the measured colocalized fraction and distances can be considered a good estimation of overlapping distributions of two molecules. We inserted in the text an additional sentence to summarize this point.